# *Ophiopogon* Polysaccharide Promotes the In Vitro Metabolism of Ophiopogonins by Human Gut Microbiota

**DOI:** 10.3390/molecules24162886

**Published:** 2019-08-08

**Authors:** Huai-You Wang, Shu-Chen Guo, Zhi-Tian Peng, Cheng Wang, Ran Duan, Tina T. X. Dong, Karl W. K. Tsim

**Affiliations:** 1Shenzhen Key Laboratory of Edible and Medicinal Bioresources, HKUST Shenzhen Research Institute, Shenzhen 518057, China; 2Division of Life Science and Center for Chinese Medicine, The Hong Kong University of Science and Technology, Hong Kong 999077, China

**Keywords:** ophiopogonin, human gut microbiota, LC-MS/MS, *Ophiopogon* polysaccharide, promotion effects

## Abstract

Gut microbiota play an important role in metabolism of intake saponins, and parallelly, the polysaccharides deriving from herbal products possess effects on gut microbiota. Ophiopogonis Radix is a common Chinese herb that is popularly used as functional food in China. Polysaccharide and steroidal saponin, e.g., ophiopogonin, mainly ophiopogonin D (Oph-D) and ophiopogonin D’ (Oph-D’), are the major constituents in this herb. In order to reveal the role of gut microbiota in metabolizing ophiopogonin, an in vitro metabolism of Oph-D and Oph-D’ by human gut microbiota, in combination with or without *Ophiopogon* polysaccharide, was conducted. A sensitive and reliable UPLC-MS/MS method was developed to simultaneously quantify Oph-D, Oph-D’ and their final metabolites, i.e., ruscogenin and diosgenin in the broth of microbiota. An elimination of Oph-D and Oph-D’ was revealed in a time-dependent manner, as well as the recognition of a parallel increase of ruscogenin and diosgenin. *Ophiopogon* polysaccharide was shown to stimulate the gut microbiota-induced metabolism of ophiopogonins. This promoting effect was further verified by increased activities of β-D-glucosidase, β-D-xylosidase, α-L-rhamnosidase and β-D-fucosidase in the broth. This study can be extended to investigate the metabolism of steroidal saponins by gut microbiota when combined with other herbal products, especially those herbs enriched with polysaccharides.

## 1. Introduction

Ophiopogonis Radix, the tuberous roots of *Ophiopogon japonicus* (Thunb.) Ker-Gawl. (Liliaceae), known as Maidong in Chinese, is a common herb being used either as a health food or a therapeutic agent for prevention and treatment of diseases [1]. The main components of Ophiopogonis Radix include steroidal saponins, homo-isoflavonoids, and polysaccharides. Pharmacologic studies have revealed the activities of Ophiopogonis Radix and its ingredients, including cardiovascular protection, immunomodulation, anti-diabetic, anti-inflammation, anti-cancer, and anti-oxidation activities [1,2,3].

Steroidal saponins are the major active constituents in Ophiopogonis Radix, having two major types: the ruscogenin-type (Rus-type) and the diosgenin-type (Dio-type). Oral pharmacokinetic study revealed the low oral bioavailability of Rus-type and Dio-type steroidal saponins [4,5]. Meanwhile, ruscogenin and diosgenin were reported to be the metabolites of Rus-type and Dio-type steroidal saponins, which were being absorbed into the blood [6,7]. Several forms of evidence have suggested the role of gut microbiota in metabolism of administered saponin glycosides, and which promotes the process of de-glycosylation [8,9]. Similar to other herbal medicines, Ophiopogonis Radix is always taken orally, and steroidal saponins within the herb are speculated to be metabolized firstly by gut microbiota before their final absorption into blood circulation [10]. Ophiopogonin D (Oph-D) and ophiopogonin D’ (Oph-D’) are the most abundant saponins in Ophiopogonis Radix belonging to Rus-type and Dio-type, respectively. Indeed, these saponins have been documented as the chemical markers for the quality control of the herb [11,12,13]. The transformation of Oph-D and Oph-D’ into their final de-glycosylated metabolites, i.e., ruscogenin and diosgenin, therefore could be triggered by fermentation together with gut microbiota.

Polysaccharide is considered as another major active constituent of Ophiopogonis Radix having up to 50% by weight [2]. The primary structures of *Ophiopogon* polysaccharides have been proposed to consist mainly of β-fructose and a small amount of α-glucose with molecular weight ranging from 2.74 × 10^3^ Da to 3.25 × 10^5^ Da [2,14,15]. *Ophiopogon* polysaccharide has been reported to possess modulation effects on gut microbiota, i.e., promoting the growth of some probiotics (e.g., *Bifidobacterium* and *Lactobacillus*) and inhibiting the accumulation of pathogenic bacteria (e.g., *Escherichia* and *Desulfovibrionaceae*), and which could serve as functional foods or prebiotics in preventing gut microbiota dysbiosis and short-chain fatty acids (SCFAs) metabolic disorder [15,16,17]. Glycosidase is mainly produced by *Bifidobacterium* spp., *Lactobacillus* spp. and *Bacteroides* spp. in gut microbiota, and plays a major role in the metabolism of saponins [18,19]. Thus, the promoting effects of *Ophiopogon* polysaccharide on *Bifidobacterium* and *Lactobacillus* could probably enhance the glycosidase activities, and further affect the metabolism of ophiopogonin.

In order to reveal the interaction of gut microbiota and *Ophiopogon* polysaccharide in ophiopogonin metabolism, human gut microbiota was fermented with Oph-D and Oph-D’, in the present of *Ophiopogon* polysaccharides. After fermentation, the elimination of ophiopogonin (Oph-D and Oph-D’) and the formation of their aglycone (ruscogenin and diosgenin) were revealed by an UPLC-MS/MS method using multiple reaction monitoring (MRM) in a positive ion mode. In addition, the increase of metabolism in *Ophiopogon* polysaccharide-treated gut microbiota was verified by assaying the enzymatic activities of β-D-glucosidase, β-D-xylosidase, α-L-rhamnosidase and β-D-fucosidase.

## 2. Results

### 2.1. Establishment of UPLC-MS/MS Analysis

The multiple reaction monitoring (MRM) in positive ion mode was identified for each ophiopogonin (i.e., Oph-D, Oph-D’, ruscogenin and diosgenin), considering sensitivity and reproducibility of dominant ions in full-scan mass spectra, as reported previously [20,21]. The 4 ophiopogonins were firstly characterized based on their mass spectra from syringe pump infusion analysis, as to determine their precursor ions and selected product ions for MRM. Then, the parameters included capillary voltage, entrance voltage, collision energy and dwell time were optimized, based on the peak areas of analytes to minimize the matrix effect, as well as to increase overall sensitivity. The optimized parameters are shown in Table 1. Chemical structure, representative MS/MS fragmentation spectrum showing precursor ion to product ion transitions and MRM chromatogram of blank broth solution spiked with each ophiopogonin at LLOQ (0.24 μM) of Oph-D, ruscogenin, Oph-D’ and diosgenin are shown in Figure 1.

To validate the developed method, a series of attributes were estimated, including calibration range, linearity, sensitivity, accuracy, precision, stability, extraction efficiency and matrix effect. The linearity of calibration curve for ophiopogonins was excellent among a concentration range of 0.24 to 60.0 μM, having the coefficient of determination (*R*^2^) greater than 0.99 (Table 2). The sensitivity was evaluated by determination of LODs and LLOQs for each ophiopogonin (Table 2). Each ophiopogonin showed a LLOQ lower than 0.24 μM, i.e., 48 fmol per test, calculated by injection volume of 2 μL and ten times dilution during the sample preparation, this value was comparable with previous determination of Oph-D, Oph-D’, ruscogenin and diosgenin in plasma [22,23].

The intra- and inter-assay performance was assessed by analyzing the 3-level of QC samples, representing the entire calibration range. For intra- and inter-assay accuracy, the relative error was -3.61% to 9.12% and −1.79% to 11.51%, respectively (Table 3). For intra- and inter-assay precision, the RSD was ranged from 3.74%–14.87% and 3.96%–13.63%, respectively. In addition, the ophiopogonins were stable for 48 h in autosampler at 4 °C by analyzing 3-level of QC samples, with a RSD below 12.64% (Table 4). 

The extracting efficiencies and matrix effects of ophiopogonins were determined in five replicates, and the results were summarized (Table 4). The extraction at three concentrations was satisfactory, as confirmed by recovery from 86.50%–104.61%. There was no significant matrix effect observed, i.e., 87.83%–97.85%, which was within the range of 80%–120% [24]. In addition, the RSD of extraction efficiencies and matrix effects for 3-level QC samples were below 15%, demonstrating a favorable sample preparation. The aforementioned parameters indicated the developed method could be used for precise determination of the chosen 4 ophiopogonins.

### 2.2. Metabolism of Ophiopogonins with Gut Microbiota

The validated method was applied to determine the concentrations of Oph-D, Oph-D’, ruscogenin and diosgenin in the fermented broth. The broth was collected from different ophiopogonin groups in the present of *Ophiopogon* polysaccharide, i.e., normal group and *Ophiopogon* polysaccharide (OJP) group (Appendix A). The concentrations of Oph-D and ruscogenin in various time points of ophiopogonin alone (normal group), as well as in ophiopogonin + *Ophiopogon* polysaccharide group (OJP group), were fully determined and compared (Appendix A). The elimination of Oph-D and formation of ruscogenin were in time-dependent manners during the fermentation, as shown in Figure 2. The concentrations of Oph-D in both groups were decreased over time by ~40% and ~64%, and therefore Oph-D was transformed into ruscogenin. In parallel, the concentrations of ruscogenin in both groups were increased over time. These results suggesting a stimulating effect of *Ophiopogon* polysaccharide on the metabolism of Oph-D, as well as the production of ruscogenin, which were further validated by calibration of AUC_0-72h_ of Oph-D and ruscogenin in each group (Appendix A).

The promoting effects of *Ophiopogon* polysaccharide in metabolizing Oph-D’ and production of diosgenin were determined (Appendix A). Most of the Oph-D’ was metabolized during the first 30 h in ophiopogonin alone (normal group) and ophiopogonin + *Ophiopogon* polysaccharide group (OJP group) (Figure 2); while the concentration of diosgenin, i.e., the final de-glycosylated metabolite of Oph-D’, in OJP group was significantly higher than that of the normal group (Figure 2). The AUC_0-72h_ of Oph-D’ and diosgenin in the normal group and OJP group were also calculated and compared; significant differences were observed for the OJP group, as compared to that of normal group (Appendix A). Thus, *Ophiopogon* polysaccharide could significantly promote the metabolism of Oph-D’ and production of diosgenin.

### 2.3. Glycosidase Activities during Fermentation with Gut Microbiota

Both Oph-D and Oph-D’ have three sugar moieties, i.e., β-D-xylose (Xyl), α-L-rhamnose (Rha) and β-D-fucose (Fuc), attached to the C-1 of ruscogenin forming Oph-D, as well as Xyl, Rha and β-D-glucose (Glc) moieties attached to the C-3 of diosgenin forming Oph-D’. Thus, the enzymes in microbiota responsible in hydrolyzing the sugars, i.e., β-D-glucosidase, β-D-xylosidase, α-L-rhamnosidase and β-D-fucosidase, were determined, as to reveal the promoting effects of *Ophiopogon* polysaccharide. The broth was collected at each time points. The release of p-nitrophenol (PNP) was reflecting the enzymatic activity in the assay. Here, the release of PNP after incubation of *p*-nitrophenyl-β-D-glucopyranoside (PNP-β-D-Glu), *p*-nitrophenyl-β-D-xylopyranoside (PNP-β-D-Xyl), *p*-nitrophenyl-α-L-rhamnopyranoside (PNP-α-L-Rha) and *p*-nitrophenyl β-D-fucopyranoside (PNP-β-D-Fuc) with the fermented broth in different groups were compared.

*Ophiopogon* polysaccharide led to significant increase of the four selected glycosidases after the early stage of fermentation at 12 h, as compared with the normal group (Figure 3). Interestingly, this phenomenon was consistent with the results as demonstrated in Figure 2, which indicated the promoting effects of *Ophiopogon* polysaccharide on metabolism of ophiopogonin D and ophiopogonin D’, as well as the production of ruscogenin and diosgenin. The activities of four glycosidases reached the highest level at about 36–60 h, and the order of activity was as follows: β-D-glucosidase > β-D-xylosidase > β-D-fucosidase > α-L-rhamnosidase. For example, at the time points of 36 h, the released PNP of 0.072 mM (PNP-β-D-Glu), 0.29 mM (PNP-β-D-Xyl), 0.014 mM (PNP-α-L-Rha) and 0.014 mM (PNP-β-D-Fuc) in normal group were increased by 71.8%, 111.0%, 51.3% and 132.0% in the polysaccharide-treated group, respectively. Thus, gut microbiota could produce the four selected glycosidase, *Ophiopogon* polysaccharide promoted the glycosidase activities of gut microbiota. 

## 3. Discussion

The LC-MS/MS assay is a common method for quantitation of ophiopogonins in biological matrix sample; however, most of the developed methods are focusing on plasma samples for pharmacokinetic study on a single ophiopogonin [20,23,25,26,27]. No reports have investigated the simultaneously quantitative analysis of ophiopogonins in fecal samples or fermented broth of gut microbiota. Here, a sensitive and reliable UPLC-MS/MS approach using multiple reaction monitoring (MRM) in positive ion mode was developed simultaneously to quantify Oph-D, Oph-D’, ruscogenin and diosgenin in the broth. The quantitative results revealed the time-profile elimination of Oph-D/Oph-D’ and formation of ruscogenin/diosgenin, and which indicated that *Ophiopogon* polysaccharide could significantly promote the metabolism of ophiopogonins.

Several in vitro models have been developed and applied for simulation the fermentation and metabolism of human colon, including batch fermentation model [9], TIM-2 model (human large intestine) [28] and SHIME model (simulator of the human microbial ecosystem) [29]. The incubation time for these in vitro models was ranged from 24 to 72 h, which corresponded to the colon retention time of in vivo study. Batch fermentation model, using general anaerobic medium (GAM) and fresh feces (the source of gut microbiota) in the present study, was the most commonly model for the in vitro metabolism study of natural products by gut microbiota, such as saponins, flavonoids, quinones, terpenoids, and alkaloids [9,30]. It has been reported that the metabolism of saponins by gut microbiota is variable between individuals, and is probably dependent on the composition of gut microbiota [31]. Therefore, 8 healthy volunteers were selected to avoid the gut microbiota diversity between individuals, and their fresh feces were collected and mixed for the preparation of human gut microbiota according to previous publication [9]. 

The transformation of triterpenoid saponins to different metabolites by gut microbiota has been well reported, de-glycosylation reactions by stepwise cleavage of the sugar moieties are the main metabolic pathway, e.g., ginsenosides and astragalosides [8,9]. However, the metabolism of steroidal saponins by gut microbiota has not been fully revealed. Rus-type and Dio-type steroidal saponins were shown to have low oral bioavailability [4,5]. In parallel, the aglycone (i.e., ruscogenin and diosgenin) were reported to be active ingredient being absorbed into blood [6,7]. Oph-D and Oph-D’ are the representative steroidal saponins belonging to Rus-type and Dio-type, respectively. Our study revealed the time-dependent metabolic process of Oph-D and Oph-D’ by gut microbiota, as promoted by *Ophiopogon* polysaccharide. These results can help us to understand the pharmacokinetics of steroidal saponins and their mechanism during common oral administration.

In addition to ruscogenin and diosgenin (the final de-glycosylated metabolites of Oph-D and Oph-D’), series numbers of inter-mediate metabolites were also detected, which were produced by stepwise cleavage of the sugar moieties in C-1 and/or C-3. For instance, ophiopogonin B (i.e., Rha and Fuc attached to C-3 of ruscogenin) and ruscogenin 1-O-β-D-fucopyranoside (i.e., Fuc attached to C-3 of ruscogenin) were detected and identified as the inter-mediate metabolites of Oph-D, as well as ophiopogonin C’ (i.e., Rha and Glu attached to C-1 of diosgenin) and diosgenin 3-O-β-D-glucopyranoside (i.e., Glu attached to C-1 of diosgenin) for Oph-D’. The stepwise cleavage of sugar moieties of Oph-D/Oph-D’ were similar to other saponins [8,9]. However, the quantitative analysis of these inter-mediate metabolites was not performed due to a lack of reference standards. The quantitative difference of reduced Oph-D/Oph-D’ and increased ruscogenin/diosgenin in Figure 2, Appendix A and Appendix A was actually the amount of the unquantified inter-mediate metabolites.

Gut microbiota can produce number of glycosidases, which releases the aglycones of saponins from their glycosides and glucuronides. The glycosidase activities can be affected by the dietary change and physiological factors, for instance, NUTRIOSE (a prebiotic fiber) and ginseng polysaccharides showed prebiotic effects on gut microbiota, enhancement effects on glycosidase activities, and promotion effects on metabolism of ginsenosides [32,33,34]. Several indigenous bacteria of the gut microbiota including *Bifidobacterium* spp., *Lactobacillus* spp. and *Bacteroides* spp. have been demonstrated to possess high glycosidases activities and involved in the metabolism of saponins [18,19]. Among them, genus *Bifidobacterium* spp. is one of the most important gut bacteria that releasing all of the four glycosidases selected in the present study, e.g., β-D-glucosidase [35], β-D-xylosidase [36], α-L-rhamnosidase [37] and β-D-fucosidase [38]. Significant difference of the glycosidase activities were observed among the four glycosidases (i.e., β-D-glucosidase > β-D-xylosidase > β-D-fucosidase > α-L-rhamnosidase), these results were consistent with previous studies, and the differences were probably caused by the composition of gut microbiota [19,39,40]. *Ophiopogon* polysaccharide has been reported to possess modulation effects on gut microbiota, i.e., promoting the growth of some probiotics (e.g., *Bifidobacterium* and *Lactobacillus*) and inhibiting the accumulation of pathogenic bacteria (e.g., *Escherichia* and *Desulfovibrionaceae*) [15,16,17]. Thus, the promoting effects of *Ophiopogon* polysaccharide on *Bifidobacterium* is probably the key factor for the enhancement of glycosidase activities and the metabolism of ophiopogonin.

## 4. Materials and Methods

### 4.1. Chemicals and Reagents

Standards of Oph-D, Oph-D’, ruscogenin, diosgenin and digoxin (Internal standard, IS) were supplied by Testing Laboratory for Chinese Medicine of Hong Kong University of Science and Technology (HKUST, Hong Kong, China). The purity of each standard was over 98%, as detected by HPLC-DAD and ^13^C-NMR analysis. The HPLC grade acetonitrile and formic acid were obtained from Merck (Darmstadt, Germany). Deionized water (18 MΩ cm^−1^) was supplied with a Direct-Q water purification system (Millipore, Milford, MA, USA). *p*-nitrophenol (PNP) and *p*-nitrophenyl-α-L-rhamnopyranoside (PNP-α-L-Rha) were purchased from Sigma-Aldrich (St. Louis, MO, USA). *p*-nitrophenyl-β-D-glucopyranoside (PNP-β-D-Glu), *p*-nitrophenyl-β-D-xylopyranoside (PNP-β-D-Xyl), and *p*-nitrophenyl β-D-fucopyranoside (PNP-β-D-Fuc) were purchased from Macklin Biochemical Co., Ltd. (Shanghai, China). The polysaccharide was isolated from Ophiopogonis Radix that was collected from Sichuan Province of China and identified by Dr. Tina Dong. The voucher samples of Ophiopogonis Radix were deposited at Shenzhen Key Laboratory of Edible and Medicinal Bioresources at HKUST. The polysaccharide was extracted in water by heat-reflux and purified by D101 macroporous resin (Shanxi Lebo Biochemical Technology, Xi’an, China), then evaporated and precipitated by 95% ethanol, and further purified by Sephadex G-25 columns (GE Healthcare Bio-Sciences AB, Uppsala, Sweden). The structural analysis showed that the *Ophiopogon* polysaccharide was homogeneous with poly-dispersity indexes of 1.3, molecular weight of 2.48 kDa, and consisted of fructose and glucose in the ratio of 17: 1: this polysaccharide was characterized in detail elsewhere [17]. Mixed solutions of Oph-D and Oph-D’ used for in vitro metabolism were freshly prepared in 75% ethanol at each concentration of 1.0 mM. 

### 4.2. Human Gut Microbiota Preparation

Fresh human feces were collected from 8 healthy Chinese volunteers (20–35 years, 4 males and 4 females) from Shenzhen, China. All volunteers were in good health and had not been given antibiotics for at least 3 months before the collection. Samples were collected and mixed, on site, on the day of the experiment and were used immediately. Fecal slurries were prepared by mixing fresh feces samples with autoclaved PBS (0.1 M, pH 7.2) to yield 10% (*w/v*) suspension. The fecal suspension was filtered with two layers of gauze, and the filtered suspension was centrifuged at 500 rpm for 5 min, then the result supernatants were collected and used for in vitro fermentation.

### 4.3. Fermentation of Ophiopogonin with Gut Microbiota

General anaerobic medium (GAM) broth was prepared according to previous publication [30], including 0.3 g of L-cysteine hydrochloride, 0.3 g of sodium thioglycolate, 1.2 g of beef liver extract powder, 2.2 g of beef extract, 2.5 g of KH_2_PO_4_, 3.0 g of soya peptone, 3.0 g of NaCl, 3.0 g of soluble starch, 5.0 g of yeast extract, 10.0 g of tryptone, 10.0 g of proteose peptone, 13.5 g of digestible serum powder, and 3.0 g of glucose were dissolved in 1,000 mL distilled water, and adjusted the pH value to 7.3. After treated with anti-bacteria process with high-pressure (0.15 MPa) and high-temperature (121 °C) for 20 min, the cooled GAM broth was then transferred to an anaerobic chamber (37 °C, anaerobic condition) and sub-packaged to sterilized vessels (50 mL volume, 30 mL of GAM per vessel). Vessels were then randomly distributed into two sub-groups (*n* = 5 for each group): normal control group (GAM broth inoculated with 3 mL of human gut microbiota and supplemented with 1 mL autoclaved PBS) and normal group (i.e., ophiopogonin group, GAM broth inoculated with 3 mL of human gut microbiota and supplemented with 1 mL mixed solutions of Oph-D and Oph-D’ (1.0 mM for each). Polysaccharide anaerobic medium (PAM) broth was prepared as that of GAM broth having *Ophiopogon* polysaccharide instead of glucose, i.e., 3.0 g polysaccharide in 1,000 mL broth. Two sub-groups (*n* = 5 for each group), using PAM broth, were also obtained as follows: *Ophiopogon* polysaccharide (OJP) control group and OJP group (i.e., OJP + ophiopogonin group). In vitro metabolism was run under an anaerobic condition at 37 °C for a period of 72 h: two parallel samples (0.5 mL for each) were collected at ten time points in sterile Eppendorf tube (1·5 mL) from normal control group, normal group, OJP control group and OJP group, one sample was used for the assay of enzyme activity immediately, and the other sample was frozen at −80 °C until for UPLC-MS/MS analysis.

### 4.4. Assay for Glycosidase Activities 

The assay for glycosidase was conducted based on the methods described in previous reports with minor modifications [31,40,41]. Briefly, 0.5 mL fermented broth of each sample collected at different time from different groups was centrifuged at 12,000 rpm for 10 min, the resulting supernatant was used for the assay of glycosidase activity. For the assay of β-D-glucosidase, β-D-xylosidase, α-L-rhamnosidase and β-D-fucosidase activities, the reaction mixture (total volume of 0.5 mL) was composed of 0.2 mL of 1 mM PNP-β-D-Glu, PNP-β-D-Xyl, PNP-α-L-Rha and PNP-β-D-Fuc as substrate respectively, 0.2 mL of 0.1 M phosphate buffer (pH 7.0), and 0.1 mL of supernatant fraction. The reaction mixture was incubated at 37 °C for 30 min. The reaction was stopped by the addition of 0.5 mL of 0.5 N NaOH, centrifuged at 3,000 rpm for 10 min and measured the absorbance at 405 nm (Microplate Spectrophotometer, BioTek Epoch 2). The absorbance of a series of different concentrations of PNP was used to calculate the enzymatic activity. In addition, an assay of glycosidase activity was also conducted for both GAM broth and PAM broth, which were used for the blank control for the normal group and the OJP group, respectively.

### 4.5. Sample Preparation for UPLC-MS/MS

Individual stock solution of Oph-D, Oph-D’, ruscogenin and diosgenin were prepared in methanol at concentration of 2.4 mM. Mixed stock solutions, containing 60.0 μM of each ophiopogonin, were made by mixing same volume of the 4 stock solutions and diluting with methanol. The calibration standard solutions (0.24–60.0 μM) and quality control (QC) solution (0.6, 3.0 and 30.0 μM) were prepared by serial dilution of mixed stock solutions with methanol. The IS stock solution (20 μM) was prepared in methanol. Fermented broth, collected at ten time points of normal control group and OJP control group, were mixed equally and used as blank broth matrix solution. The blank broth matrix solution (100 μL), IS stock solution (50 μL), calibration standard solution (100 μL) or QC solution (100 μL) were mixed together and added up to 1.0 mL with methanol. The result solutions were used as calibration sample or QC sample, respectively. Fermented broth of each sample (100 μL) and IS stock solution (50 μL) were mixed together, and then added up to 1.0 mL with methanol, the resulted solution was used as test sample. Each sample was vortexed for 1 min and centrifuged at 12,000 rpm for 10 min. A volume of 2 μL aliquot of each supernatant was injected into UPLC-MS/MS for analysis.

### 4.6. UPLC-MS/MS Analysis

Chromatographic analysis was performed on an A30 Altus™ UPLC system (PerkinElmer, Waltham, MA, USA). The autosampler was kept at 4 °C. Sample separation was achieved on an ACQUITY UPLC BEH C_18_ column (2.1 mm × 50 mm, 1.7 μm) (Waters, Milford, MA, USA) with a constant flow rate of 0.4 mL/min at 30 °C. The mobile phase was composed of water (0.1% formic acid, A) and acetonitrile (0.1% formic acid, B), using a gradient elution of 40%–60% B at 0–3 min, 60%–90% B at 3–13 min, 90% B at 13–15 min, and maintained at 40% mobile phase B for an additional 2 min for re-equilibration. The injected volume was set at 2 μL. Detection was performed by Qsight™ 220 triple quadrupole mass spectrometer (PerkinElmer), using multiple reaction monitoring (MRM) in the positive ion mode. The acquired parameters were optimized as follows: drying gas value, 100; nebulizer gas value, 180; electrospray voltage, 5500 V; HSID temperature, 280 °C. 

### 4.7. Method Validation

The linearity, sensitivity, precision, accuracy, stability, matrix effect and extraction efficiency were determined to evaluate the integrity of the developed method according to the criteria described in the FDA guidelines for bioanalytical method validation [42]. The calibration was conducted based on the analysis of calibration standards. Linearity was assessed by establishing the calibration curves by plotting the peak response ratios (Y) of each analyte versus IS against their corresponding concentrations (X, μM), and were fitted by linear regression with a weight factor of 1/X. Analytical sensitivity was determined as limit of detection (LOD) and lower limit of quantitation (LLOQ) of each ophiopogonin, which were estimated at signal-to-noise (*S/N*) ratios of 3: 1 and 10: 1, respectively. Five replicates were performed at each point. Intra- and inter-assay precision and accuracy were assessed by analyzing QC samples at three different levels. The intra-assay accuracy and precision were estimated using relative error (RE) and relative standard deviation (RSD) of selected QC concentration within a run (*n* = 5), respectively. Inter-assay accuracy and precision were determined on three consecutive days (*n* = 15). Short-term autosampler stability of each ophiopogonin was evaluated by analysis of the QC samples at 8 h intervals over 48 h (4 °C) and calculated by recoveries and relative standard deviation (RSD).

The extraction efficiency of each ophiopogonin from fermented broth was calculated by comparing the peak areas ratios of the analytes to the IS of pretreated QC samples (low, medium and high concentration) with those of post-extracted blank broth, spiked with the ophiopogonin at the same concentration. The matrix effect was assessed by comparing peak areas ratios of the analytes to the IS of post-extracted spiked samples with that of standards, lacking broth matrix at QC levels. Five replicates for measurement of extraction efficiency and matrix effect were performed.

### 4.8. Data Analysis

Simplicity 3Q™ software platform (PerkinElmer, Waltham, MA, USA) was used for data acquisition and processing. The concentrations of each ophiopogonin were calculated according to the calibration curve. The area under the concentration–time curve (AUC_0-72h_) were analyzed using Graphpad Prism software (v7.0). Data were expressed as mean ± standard deviation (SD) (*n* = 5). The significance of differences in data between the groups was determined by one-way ANOVA with Duncan’s test. 

## 5. Conclusions

The in vitro metabolism of ophiopogonins (Oph-D and Oph-D’) into their aglycone (ruscogenin and diosgenin) in combining with or without *Ophiopogon* polysaccharide was conducted. The quantitative results revealed the time-profile elimination of ophiopogonins and formation of their metabolites, indicating that *Ophiopogon* polysaccharide could significantly promote the metabolism of ophiopogonins. The promoting effect of *Ophiopogon* polysaccharide in metabolism of ophiopogonins was further validated by the increased enzymatic activities of β-D-glucosidase, β-D-xylosidase, α-L-rhamnosidase and β-D-fucosidase. Thus, our study can be extended to investigate the metabolism of steroidal saponins by gut microbiota when combined with other nature products.

## Figures and Tables

**Figure 1 molecules-24-02886-f001:**
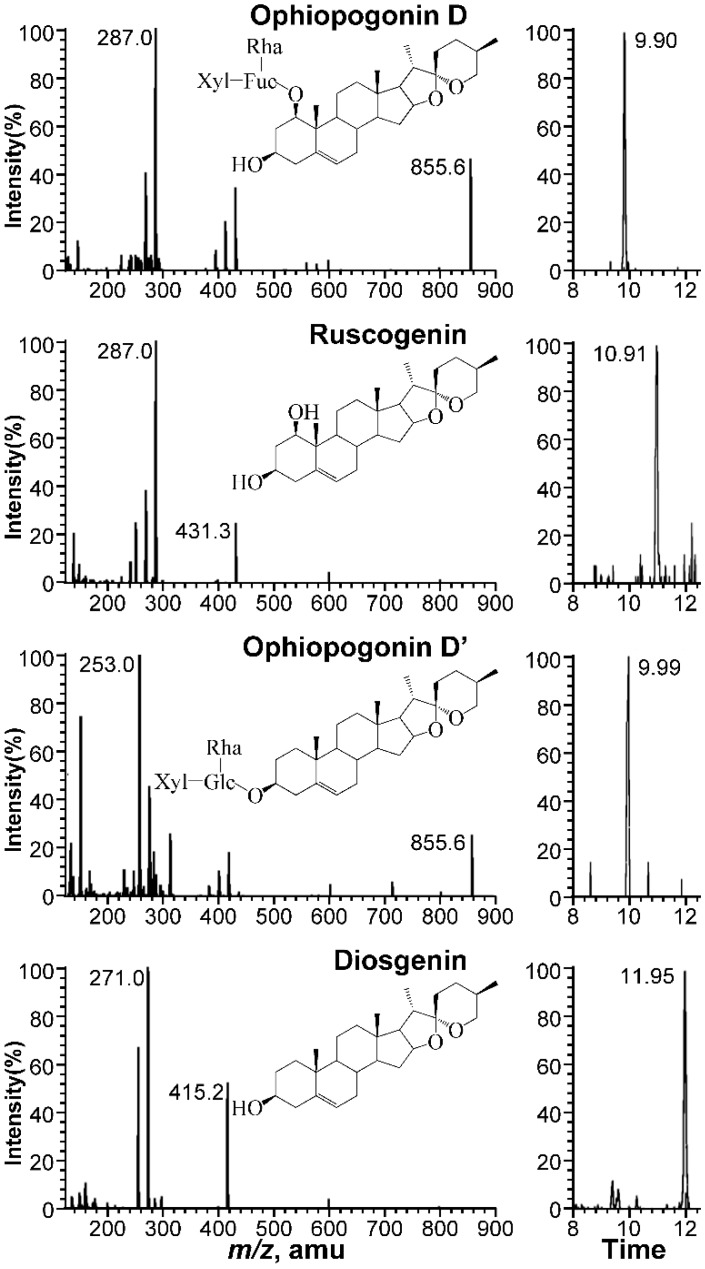
Chemical structure, representative MS/MS fragmentation spectrum showing precursor ion to product ion transitions, MRM chromatogram of blank broth solution spiked with each ophiopogonin at LLOQ (0.24 μM) of ophiopogonin D, ruscogenin, ophiopogonin D’ and diosgenin.

**Figure 2 molecules-24-02886-f002:**
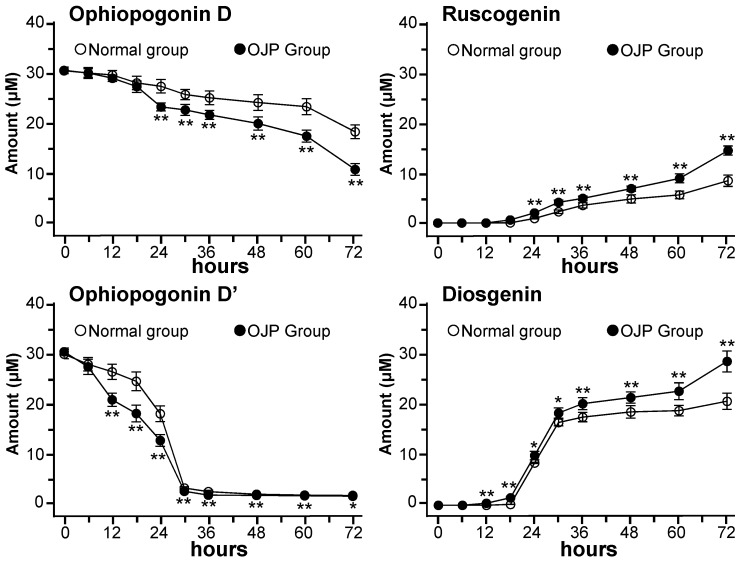
*Ophiopogon* polysaccharide promotes the metabolism of ophiopogonin D and ophiopogonin D’, as well as the production of ruscogenin and diosgenin. Time-profile elimination/formation of ophiopogonin D, ruscogenin, ophiopogonin D’ and diosgenin in different conditions. Normal: ophiopogonin group; OJP: *Ophiopogon* polysaccharide (OJP) + ophiopogonin group. Data are represented as mean ± SD (*n* = 5). Significance difference was assessed by one-way ANOVA: * *p* < 0.05 and ** *p* < 0.01 vs. normal group.

**Figure 3 molecules-24-02886-f003:**
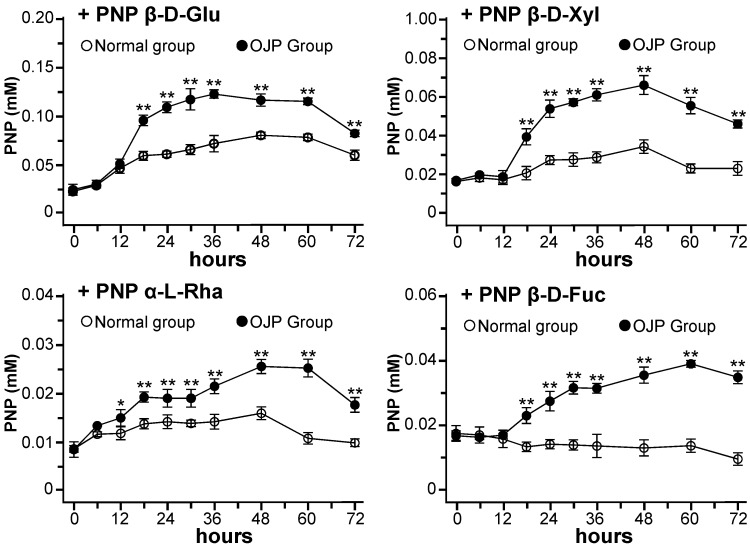
*Ophiopogon* polysaccharide promote the glycosidase activities of human gut microbiota. Release of PNP after incubation of PNP β-D-Glu, PNP β-D-Xyl, PNP α-L-Rha and PNP β-D-Fuc with human gut microbiota in different groups were measured in each time points. Normal: ophiopogonin group; OJP: *Ophiopogon* polysaccharide (OJP) + ophiopogonin group. PNP: *p*-nitrophenol; PNP β-D-Glu: *p*-nitrophenyl-β-D-glucopyranoside; PNP β-D-Xyl: *p*-nitrophenyl-β-D-xylopyranoside; PNP α-L-Rha: p-nitrophenyl-α-L-rhamnopyranoside; PNP β-D-Fuc: *p*-nitrophenyl β-D-fucopyranoside. Data are represented as mean ± SD (*n* = 5). Significance difference was assessed by one-way ANOVA: * *p* < 0.05 and ** *p* < 0.01 vs. the normal group.

**Table 1 molecules-24-02886-t001:** Analytic parameters of ophiopogonins by UPLC-MS/MS in fermented broth of human gut microbiota.

No.	Rt (min)	Analyte	Mw	Precursor Ion	Product Ion	Entrance Voltage (V)	Collision Energy (eV)
1	9.90	Ophiopogonin D	855.02	855.6	287.0	20	−55
2	9.99	Ophiopogonin D’	855.02	855.6	253.0	20	−72
3	10.91	Ruscogenin	430.62	431.3	287.0	20	−24
4	11.95	Diosgenin	414.62	415.2	271.0	20	−23
IS	4.23	Digoxin	780.94	803.5	387.0	20	−26

**Table 2 molecules-24-02886-t002:** Linearity and sensitivity of UPLC-MS/MS assay for ophiopogonins in fermented broth of human gut microbiota.

Analyte	Calibration Curve	*R* ^2^ ^a^	Linear Range (μM)	LLOQ ^b^ (μM)	LOD ^c^ (μM)
Ophiopogonin D	y = 0.4461x − 0.0870	0.9986	0.24–60	0.24	0.07
Ophiopogonin D’	y = 0.4688x − 0.0442	0.9976	0.24–60	0.24	0.07
Ruscogenin	y = 0.3848x + 0.1414	0.9985	0.24–60	0.24	0.07
Diosgenin	y = 0.3767x + 0.1790	0.9977	0.24–60	0.24	0.07

^a^*R*^2^, coefficient of determination; ^b^ LLOQ: lower limit of quantification; ^c^ LOD: limit of detection.

**Table 3 molecules-24-02886-t003:** The intra-day and inter-day precision and accuracy for UPLC-MS/MS assay of ophiopogonins in fermented broth of human gut microbiota.

Analyte	Spiked (μM)	Intra-Day (*n* = 5)			Inter-Day (*n* = 15)		
		Measured (μM)	Precision (RSD%)	Accuracy (RE%)	Measured (μM)	Precision (RSD%)	Accuracy (RE%)
Ophiopogonin D	0.60	0.62	3.80	3.43	0.59	11.87	−1.79
	3.00	2.89	8.09	−3.61	3.06	13.63	2.10
	30.0	29.2	10.07	−2.57	31.6	10.31	5.43
Ophiopogonin D’	0.6	0.59	12.05	−2.25	0.65	9.46	8.79
	3.0	3.11	12.88	3.60	3.24	11.51	7.91
	30.0	30.2	14.87	0.68	31.9	8.09	6.30
Ruscogenin	0.6	0.65	5.95	9.12	0.65	3.96	7.94
	3.0	3.20	4.94	6.68	3.20	6.36	6.91
	30.0	31.8	12.39	6.17	33.4	12.67	11.51
Diosgenin	0.6	0.65	6.14	8.95	0.62	8.68	4.51
	3.0	3.13	3.74	4.43	3.07	8.78	2.40
	30.0	30.19	8.04	0.62	31.6	8.45	5.19

**Table 4 molecules-24-02886-t004:** Extraction efficiency, matrix effect and stability for UPLC-MS/MS assay of ophiopogonins in fermented broth of human gut microbiota.

Analyte	Spiked (μM)	Extraction Efficiency (*n* = 5)		Matrix Effects (*n* = 5)		Autosampler Stability (*n* = 6)	
		Measured (%)	RSD (%)	Measured (%)	RSD (%)	Measured (%)	RSD (%)
Ophiopogonin D	0.60	94.74	7.85	97.85	9.85	102.91	8.58
	3.00	96.17	6.36	87.83	9.08	94.33	5.54
	30.0	104.61	13.05	89.92	5.81	99.13	12.64
Ophiopogonin D’	0.60	88.86	8.75	96.78	12.76	89.69	7.36
	3.00	96.13	10.97	94.36	6.5	97.15	8.17
	30.0	98.66	10.34	96.89	6.95	101.59	11.39
Ruscogenin	0.60	86.79	5.27	87.93	6.96	102.68	4.87
	3.00	94.54	5.7	92.28	6.98	96.95	6.67
	30.0	99.32	3.46	97.06	6.11	94.70	6.58
Diosgenin	0.60	90.17	9.08	92.29	7.89	102.60	6.52
	3.00	86.50	5.72	94.90	7.39	97.04	3.65
	30.0	96.76	4.19	97.10	5.74	92.17	4.52

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
