# Peer review of "Ophiopogon Polysaccharide Promotes the In Vitro Metabolism of Ophiopogonins by Human Gut Microbiota"

_molecules, 2019, doi:10.3390/molecules24162886_

Round 1
Reviewer 1 Report
The manuscript molecules-560719 entitled “Ophiopogon polysaccharide promotes the metabolism of ophiopogonins by human gut microbiota” by Huai-You Wang et al. is focus on the role of gut microbiota in metabolizing ophiopogonin, an in vitro metabolism of Oph-D and Oph-D’ by human gut microbiota, in combination with or without Ophiopogon polysaccharide.
I have a few concerns about the present manuscript:
-In general, the main idea of the investigation is good, but the authors have avoided a discussion section, this section is missing in the text
-The introduction need some information about the impact of ophiopogon administration in gut microbiota
-Why the authors have used fresh human feces were collected from eight healthy Chinese volunteers (20-35 years, 4 males and 4 females) from Shenzhen, China?. Eight samples are enough
-The objective of the manuscript was the role in human gut microbiota, which bacteria are represented in the difference of glycosidase activities?, please add information
Please complete the “Sample Availability: Samples of the compounds ...... are available from the authors.” in the text, information is missing.
Line 58, please define SCFA
Reviewer 2 Report
Authors present a UPLC-MS/MS method to determine components of Ophiopogonis Radix and its metabolites and in vitro metabolism with or without Ophiopogon polysaccharide in human gut microbiota. This method could be very useful for monitoring these compounds in vitro samples and applied in vivo samples. However, there are some comments should be further explained before consideration of publication:
1. Volume unit ‘ml and mL’ are mixed. Units need to be unified.
2. It would be nice to show MRM chromatograms of LLOQ or LOD. And also please explain S/N of analytes.
3. For this method validation, short-term stability should be explained.
4. The amount of reduced Ophiopogonin D (or Ophiopogonin D') (uM or AUC) and the amount of increased Ruscogenin (or Diosgenin) (uM or AUC) are different. Why?
5. The decreasing rate of Ophiopogonin D' and the increasing rate of Diosgenin are different. Why?
6. Why does Ophiopogon polysaccharide increase gut metabolism of Diosgenins? What is the propoased mechanism?
7. Transit time in small intestine or colon is about 4 or 10 hrs, respectively. The amount of change during this period (< 12 hr) is small. In this regard, an explanation of the interrelation with the results is needed.
8. The amount and rate of PNP production differ by type of glucose. Why?
Reviewer 3 Report
This study investigated effect of ophiopogon polysaccharide on ophiopogonin metabolism by intestinal bacteria. Ophiopogon polysaccaride promoted bioconvertion of ophiopogonins to their aglycone forms. This may be because of the up-regulation of glycosidase expression. However, the increase of the aglycones after 30 h incubation seems to really do not have any pharmacological and physiological meaning, because digested components generally stay in colon for around 12-20 h. It should be fully discussed. In addition, effects of other types of dietary fibers on the saponin bioconversion should be investigate to reveal whether the increase of digestive enzymes is unique action to ophiopogon polysaccaride.
Round 2
Reviewer 1 Report
The authors have addressed all my concerns. Thank you for the revised version of your manuscript.
Reviewer 3 Report
Thank you for your responses. Revised manuscript seems to be appropriate to be published in this journal.